# Release-dependent feedback inhibition by a presynaptically localized ligand-gated anion channel

Seika Takayanagi-Kiya[1†], Keming Zhou[1,2†], Yishi Jin[1,2*]

[1]Section of Neurobiology, Division of Biological Sciences, University of California, San Diego, San Diego, United States; [2]Howard Hughes Medical Institute, University of California, San Diego, San Diego, United States

**Abstract** Presynaptic ligand-gated ion channels (LGICs) have long been proposed to affect neurotransmitter release and to tune the neural circuit activity. However, the understanding of their in vivo physiological action remains limited, partly due to the complexity in channel types and scarcity of genetic models. Here we report that *C. elegans* LGC-46, a member of the Cys-loop acetylcholine (ACh)-gated chloride (ACC) channel family, localizes to presynaptic terminals of cholinergic motor neurons and regulates synaptic vesicle (SV) release kinetics upon evoked release of acetylcholine. Loss of *lgc-46* prolongs evoked release, without altering spontaneous activity. Conversely, a gain-of-function mutation of *lgc-46* shortens evoked release to reduce synaptic transmission. This inhibition of presynaptic release requires the anion selectivity of LGC-46, and can ameliorate cholinergic over-excitation in a *C. elegans* model of excitation-inhibition imbalance. These data demonstrate a novel mechanism of presynaptic negative feedback in which an anion-selective LGIC acts as an auto-receptor to inhibit SV release.

*For correspondence: yijin@ucsd.edu

†These authors contributed equally to this work

Competing interests: The authors declare that no competing interests exist.

## Introduction

Cys-loop Ligand-gated ion channels (LGICs) participate in rapid regulation of neurotransmission by changing the plasma membrane conductance upon ligand binding (*Keramidas et al., 2004*). While LGICs at postsynaptic sites have been extensively studied in many neural circuits, our knowledge for LGICs acting at the presynaptic terminals remains fragmentary (*Engelman and MacDermott, 2004*; *Rudomin and Schmidt, 1999*). For example, classical electrophysiological recordings from cat spinal cord showed that activation of flexor afferent neurons caused decreased excitatory transmission of extensor afferent neurons onto extensor motor neurons (*Eccles et al., 1961*, *1963*; *Frank and Fuortes, 1957*). This effect was blocked by picrotoxin, an antagonist of ionotropic GABA receptors, implying that GABA_A receptors may be involved in inhibition of presynaptic release (*Eccles et al., 1963*). Ultrastructural studies later revealed that the extensor afferent neuron terminals receive axo-axonal connections from GABAergic interneurons that are activated by flexor afferent neurons (*Pierce and Mendell, 1993*; *Destombes et al., 1996*), suggesting the presence of GABA-gated LGICs in presynaptic axons. Electrophysiological recordings from crayfish opener muscles, which are innervated by excitatory and inhibitory motor neurons, also provided some evidence for the function of presynaptic LGICs. It was shown that stimulation of inhibitory motor neurons caused decreased neurotransmission from excitatory motor neurons, and that statistical analysis of excitatory transmitter quanta suggested a likely presynaptic effect (*Dudel and Kuffler, 1961*). Addition of exogenous GABA at these neuromuscular junctions caused similar effects, which was relieved by picrotoxin (*Takeuchi and Takeuchi, 1966*, *1969*), suggesting this presynaptic inhibition involves GABA-gated LGICs in the presynaptic axons. Studies in the past decades have subsequently identified several

ionotropic glycine receptors and GABA$_A$ receptors acting in the presynaptic cells to regulate synaptic transmission in the mammalian central nervous system (*Kullmann et al., 2005*; *Engelman and MacDermott, 2004*). However, despite substantial molecular characterizations, there remains a large gap in our knowledge concerning the type and the subcellular specificity of presynaptic LGICs with respect to their physiological roles in tuning neuronal communication.

In *C. elegans* locomotor circuit, acetylcholine provides major excitatory inputs to the body wall muscles (*Richmond and Jorgensen, 1999*). Studies of cholinergic transmission at the neuromuscular junctions have identified conserved molecules and elucidated mechanisms regulating fast neurotransmitter release (*Rand and Nonet, 1997*; *Richmond, 2005*). Upon Ca$^{2+}$ influx under stimulation, docked synaptic vesicles fuse with the plasma membrane through rapid action involving synaptotagmins, soluble N-ethylmaleimide–sensitive factor attachment protein receptor (SNARE) proteins, and voltage-gated calcium channels. *C. elegans* genome encodes a large number of LGICs, many of which are known or predicted to be responsive to acetylcholine (*Hobert, 2013*; *Jones and Sattelle, 2008*), but whether they regulate SV release at presynaptic terminals is not known. Here, we report in vivo evidence for the ligand-gated anion channel LGC-46 (Ligand Gated ion Channel -46) in regulating SV release kinetics as a presynaptic auto-receptor. LGC-46 localizes close to SV release sites in cholinergic motor neurons. Loss of *lgc-46* specifically affects the decay phase, but not the rise phase, of evoked release to cause prolonged neurotransmitter release, without altering endogenous neuronal activity. Gain-of-function of LGC-46, caused by a missense mutation in the pore-lining transmembrane domain, accelerates decay kinetics to dampen synaptic transmission. We further show that LGC-46-mediated presynaptic inhibition ameliorates the seizure phenotype of a *C. elegans* mutant that displays excitation-inhibition imbalance in the locomotor circuit. Our results reveal a mechanism by which a presynaptic terminal-localized anion LGIC acts in a rapid feedback inhibition of neural activity.

## Results

### LGC-46 localizes to presynaptic terminals of cholinergic motor neurons

*lgc-46* is a member of the acetylcholine gated chloride (ACC) channel subfamily in *C. elegans* (*Figure 1—figure supplement 1A–C*) (*Putrenko et al., 2005*). ACC channels are Cys-loop LGICs, and their pore-lining transmembrane region of this family has extensive homology to GABA- and glycine-gated chloride channels (*Figure 1A*). Members of the family show selective response to acetylcholine and conduct anions when expressed in Xenopus oocytes (*Putrenko et al., 2005*; *Ringstad et al., 2009*).

To dissect the function of *lgc-46*, we first determined its cellular expression using a transcriptional reporter, and observed a broad expression in the nervous system, including motor neurons (*Figure 1—figure supplement 1D*). Co-expression of P*lgc-46-GFP* with markers for cholinergic (P*unc-17-mCherry*) and GABAergic (P*ttr-39-mCherry*) neurons revealed that *lgc-46* was strongly expressed in cholinergic motor neurons, and weakly expressed in GABAergic motor neurons (*Figure 1—figure supplement 1E*). To examine the subcellular localization of LGC-46 in cholinergic motor neurons, we generated functional LGC-46::GFP, in which GFP was inserted in-frame in the cytoplasmic loop between TM3 and TM4 (*Figure 1—figure supplement 1A*, and later), and expressed the transgene as a single-copy insertion (*Supplementary file 1*). In the first larval (L1) stage animals, cholinergic motor neurons form synapses onto dorsal body wall muscles (*Sulston, 1976*; *White et al., 1978*), and we observed LGC-46::GFP only in the dorsal nerve cords (*Figure 1B*). We also expressed LGC-46::GFP in a single cholinergic motor neuron DA9, which synapses onto dorsal body wall muscles, and found LGC-46::GFP was detected only in the dorsal side of L4 and adult animals (*Figure 1C*). These results indicate that LGC-46 primarily localizes to the axonal compartment of cholinergic motor neurons.

To determine if the punctate pattern of LGC-46::GFP corresponded to presynaptic terminals, we co-immunostained LGC-46::GFP together with endogenous active zone protein RIM/UNC-10 (*Figure 1D*). LGC-46::GFP displayed a high degree of overlap with UNC-10. Quantitative analyses of colocalization indicate that LGC-46 localizes to the release sites of synaptic vesicles (SVs) (*Figure 1F*). Additionally, we found that LGC-46::GFP showed co-localization with a synaptic vesicle

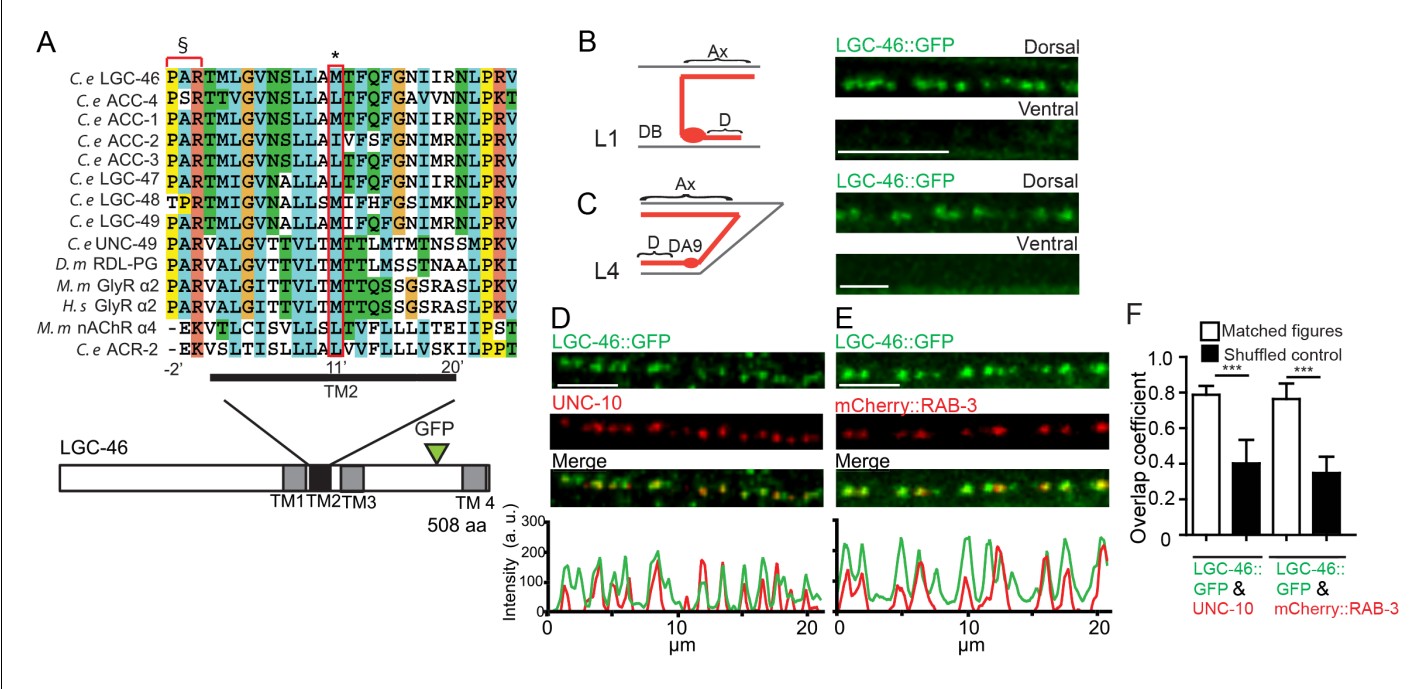

**Figure 1.** LGIC LGC-46 localizes to presynaptic terminals. (**A**) Alignment of TM2 of LGC-46 with other anion channels. * marks Met314, which is mutated to Ile in LGC-46(*ju825*). § marks PAR motif, a signature of ionotropic anion channels. (**B–F**) LGC-46::GFP localizes to presynaptic terminals of cholinergic motor neurons. (**B**) Left panel shows a schematic of a cholinergic motor neuron in L1 animals. Right panel shows a confocal image of LGC-46::GFP from P*unc-17β-LGC-46(WT)::GFP(juSi295)IV* showing punctate localization in the dorsal nerve cord. (**C**) Left panel shows a schematic of DA9 cholinergic motor neuron in the tail of L4 animals. Right panel shows LGC-46::GFP expressed in DA9 neuron, from P*itr-1-LGC-46(WT)::GFP(juEx6843),* showing punctate localization in the dorsal axon. (**B, C**): Ax: Axon. D: Dendrite. (**D, E**) LGC-46::GFP colocalizes with active zone protein UNC-10/RIM and synaptic vesicles in cholinergic motor neurons. Images of dorsal nerve cord are shown above, linescan analyses of fluorescent signal intensities below. (**D**) Conformal images of anti-GFP for LGC-46 (green) and anti-UNC-10 (red) from an animal carrying P*unc-17β-LGC-46(WT)::GFP(juSi295)IV*. (**E**) Presynaptic protein mCherry::RAB-3 expressed in cholinergic motor neurons overlapped with LGC-46::GFP signals. Images are from P*unc-17β-LGC-46 (WT)::GFP(juSi295)IV*; P*acr-2-mCherry::RAB-3(juEx7053)*. Scale bar: 5 μm. (**F**) Mander's overlap coefficient showing the extent of pixel colocalization of LGC-46 and preaynaptic marker protein signals (open bar). As negative controls, Mander's overlap coefficient from shuffled images are shown (Filled bar). n = 10 per genotype. Data shown as mean ± SD. Statistics: one way ANOVA followed by Tukey's post-hoc test. ***p<0.001.

The following figure supplement is available for figure 1:

**Figure supplement 1.** *lgc-46* is expressed in the nervous system including cholinergic motor neurons.

marker mCherry::RAB-3 expressed in cholinergic motor neurons (*Figure 1E–F*). To our knowledge LGC-46 is the first reported LGIC localized to presynaptic terminals in *C. elegans*.

## Loss of LGC-46 function prolongs synaptic transmission dependent on evoked acetylcholine release

To analyze the function of *lgc-46*, we examined two genetic deletion alleles *ok2900* and *ok2949*, both predicted to result in non-functional proteins (*Figure 1—figure supplement 1A*), and hereafter designated *lgc-46(0)*. Cholinergic synapse morphology and density in *lgc-46(0)* mutants were comparable to those in wild type (*Figure 2—figure supplement 1A–B*), and *lgc-46(0)* animals displayed normal growth, reproduction and locomotion. We assessed cholinergic transmission at the neuromuscular junction using pharmacological assays (*Mahoney and Luo, 2006*). *lgc-46(0)* animals were hypersensitive to aldicarb, an inhibitor of acetylcholine esterase, while they showed normal sensitivity to levamisole, an agonist for the postsynaptic acetylcholine receptors on muscles (*Figure 3—figure supplement 1A–B*). The hypersensitivity to aldicarb was rescued by specific expression of LGC-46 in cholinergic motor neurons (*Figure 3—figure supplement 1A*), suggesting that LGC-46 likely represses cholinergic motor neuron activities.

To directly examine how LGC-46 regulates synaptic vesicle release, we next performed electro-physiological recordings on muscle cells (*Figure 2*). Evoked excitatory post-synaptic currents (eEPSCs) represent depolarization-triggered simultaneous release of hundreds of SVs, and the kinetics of the rise and decay phase of eEPSC reflect the onset and duration of SV release, respectively. We found that the amplitude of eEPSCs in *lgc-46(0)* animals was comparable to that in wild type (*Figure 2A,C*). The rise phase of eEPSCs was similar to wild type, suggesting the onset of evoked cholinergic transmission is not affected (*Figure 2B,D*). However, the analysis of release kinetics revealed that eEPSCs in lgc-*46(0)* were prolonged and displayed a significantly slower decay than that of wild type. We calculated the cumulative charge transfer during SV release. In wild type, the cumulative charge transfer during evoked release of SVs showed a burst within 15 ms after stimulation, and a slower sustained increase afterwards. The slower decay of eEPSCs in *lgc-46(0)* mutants led to a significant increase in cumulative charge transfer (*Figure 2E*). Specifically, *lgc-46(0)* mutants displayed a normal early phase of SV release (21.33 ± 2.50 pC, compared to 18.41 ± 1.68 pC in wild type), but exhibited a large increase in the late phase (18.06 ± 2.19 pC, compared to 11.35 ± 1.35 pC in wild type). Both the slow decay of eEPSCs and the increased late phase of SV release were rescued by expression of LGC-46 in cholinergic motor neurons (*Figure 2A–E*).

To distinguish the presynaptic or postsynaptic contribution underlying the kinetic change, we recorded endogenous EPSCs in muscle cells, triggered by the spontaneous SV release from cholinergic motor neurons. The frequency of endogenous EPSCs in *lgc-46(0)* mutants was comparable to that in wild type (*Figure 2F*), indicating that cholinergic transmission in the resting condition was not affected. The amplitude and kinetics of endogenous EPSCs were normal in *lgc-46(0)* mutants (*Figure 2F*; *Figure 2—figure supplement 1C–D*), confirming that the slow decay of evoked release was not due to altered kinetics of postsynaptic receptor response on muscles. As LGC-46 resides in the presynaptic terminals of cholinergic motor neurons, these data support a conclusion that LGC-46 acts as an auto-receptor to inhibit presynaptic activities upon synaptic vesicle release.

## A missense mutation in TM2 of LGC-46 causes gain-of-function and ameliorates cholinergic over-excitation

Cholinergic motor neurons express the ACR-2 ACh-gated cation channel in their dendrites and soma (*Jospin et al., 2009*; *Qi et al., 2013*). We previously characterized an *acr-2(gf)* mutation that contains a missense mutation in the transmembrane domain 2, similar to those found in CHNR2B of human epilepsy patients (*Phillips et al., 2001*). ACR-2(gf) increases channel activity and causes hyperactivity of cholinergic motor neurons. *acr-2(gf)* animals show distinctive spontaneous convulsions, due to an altered ratio of excitation and inhibition in the locomotor circuit (*Jospin et al., 2009*; *Qi et al., 2013*; *Stawicki et al., 2013*). By screening for suppressors of the *acr-2(gf)* convulsion behavior (see Materials and methods), we isolated a *lgc-46(ju825)* mutation that strongly suppressed the convulsion of *acr-2(gf)* (*Figure 3A*). We determined that the *lgc-46(ju825)* mutation causes a substitution of methionine to isoleucine (M314I) in the pore-lining transmembrane TM2 domain (*Figure 1A*). *lgc-46(0)* did not show significant effects on *acr-2(gf)* convulsion (*Figure 3A*). In addition, transgenic overexpression of LGC-46(M314I), but not LGC-46(WT), driven by *lgc-46* promoter strongly suppressed *acr-2(gf)* convulsion. These data indicate that *lgc-46(ju825)* is a gain-of-function mutation, designated as *lgc-46(gf)* hereafter.

By expressing *lgc-46* in a cell-type specific manner, we found that pan-neuronal and cholinergic LGC-46(M314I) expression suppressed *acr-2(gf)* convulsions whereas its expression in muscles or GABAergic motor neurons had no effect (*Figure 3B*). *acr-2(gf)* shows hypersensitivity to both aldicarb and levamisole (*Jospin et al., 2009*). *lgc-46(ju825)* partially suppressed the aldicarb hypersensitivity, but not the levamisole sensitivity of *acr-2(gf)* (*Figure 3—figure supplement 1C–D*), suggesting that *lgc-46(ju825)* likely affects presynaptic activity of cholinergic neurons. Consistently, expression of LGC-46(M314I) in cholinergic motor neurons also fully suppressed the aldicarb hypersensitivity of *lgc-46(0)* (*Figure 3—figure supplement 1A*). These data show that *lgc-46* acts cell-autonomously to suppress cholinergic neuron activities.

LGC-46 is predicted to be an anion-conducting channel, and the presence of the PAR motif preceding TM2 is crucial for ion selectivity (*Galzi et al., 1992*; *Keramidas et al., 2000*; *Jensen et al., 2005*). To address if anion selectivity of LGC-46(M314I) is required for its activity, we generated mutations in the PAR motif by deleting P301 and replacing A302 with glutamate (*Figure 3C*). When

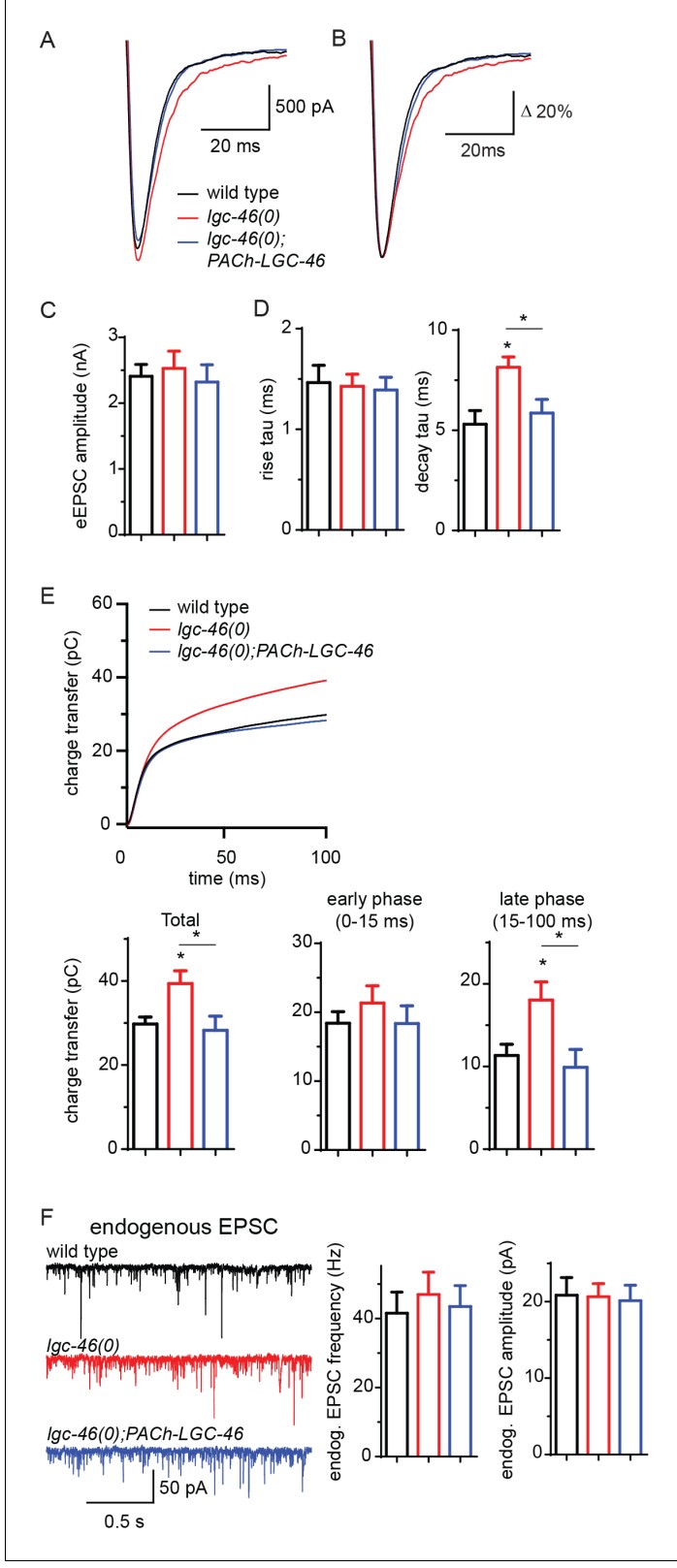

**Figure 2.** LGC-46 regulates the decay phase of eEPSCs to modulate the late phase of SV release. (**A** and **B**) Average traces (**A**) and normalized average traces (**B**) of eEPSCs. (**C**) Mean amplitude of eEPSCs. (**D**) Rise tau and decay tau of eEPSCs. (**E**) Average traces of cumulative charges of eEPSCs, and charge transfers after stimulation. (**F**) Representative traces, mean frequency and mean amplitude of endogenous EPSCs. wild type (n = 11), *lgc-46*

*Figure 2 continued on next page*

*Figure 2 continued*

(ok2900) (n = 10), and *lgc-46(ok2900); PACh-LGC-46* (n = 11). Animals were recorded at 20 degrees in 1.2 mM Ca$^{2+}$ bath solutions. Statistics, one-way ANOVA, Bonferroni's post hoc test. *p<0.05. Error bars indicate SEM.
The following figure supplement is available for figure 2:

**Figure supplement 1.** Morphology of cholinergic synapses as well as the amplitude and kinetics of endogenous EPSCs are normal in *lgc-46(0)*.

expressed in cholinergic motor neurons, PAR-mutant LGC-46(M314I) did not suppress *acr-2(gf)* convulsions (*Figure 3C*), consistent with LGC-46 acting as an anion-selective channel.

## *lgc-46(gf)* accelerates the decay phase of synaptic transmission

The M314I mutation does not alter LGC-46 localization as functional GFP-tagged LGC-46(M314I) showed restricted expression at presynaptic terminals, similar to wild type (). We next investigated how LGC-46(M314I) alters the presynaptic release by electrophysiological recordings. *lgc-46(gf)* mutants showed significantly reduced amplitudes of eEPSCs and release kinetics with much shorter decay phase compared to wild type, whereas the time constant of the rise phase of eEPSCs was not affected (*Figure 4A–D*). Consequently, the charge transfer during eEPSCs was significantly reduced in *lgc-46(gf)* mutants, indicating the inhibition of SV release (*Figure 4E*). We also recorded endogenous release and found that, similar to *lgc-46(0)*, the amplitude and kinetics of endogenous EPSCs were not affected in *lgc-46(gf)* mutants (*Figure 4F*; *Figure 4—figure supplement 1G–H*). However, *lgc-46(gf)* significantly reduced the frequency of endogenous EPSCs, suggesting the lower excitability in cholinergic motor neurons compared to wild type (*Figure 4F*). Expression of LGC-46(M314I) in *lgc-46(0)* mutant background mimicked the characteristics of reduced amplitude and shortened decay of evoked release from the *lgc-46(gf)* animals (*Figure 4A–F*). These results suggest that LGC-46(M314I) can function as an overactive anion channel, thereby suppressing SV release more efficiently than wild type LGC-46.

## LGC-46 function requires ACC-4 that also localizes to presynaptic terminals

Some ACC family proteins can form homomeric or heteromeric channels, which respond to acetylcholine when expressed in Xenopus oocytes (*Putrenko et al., 2005*). Similar studies of LGC-46 have not yet revealed ACh-gated channel activity (Joseph Dent, personal communication), or responsiveness to other neurotransmitters (*Ringstad et al., 2009*), suggesting that LGC-46 may require additional subunits or accessory factors to form a functional channel. To identify potential interacting partners, we took advantage of a behavioral phenotype of *lgc-46(gf)*. These animals display a curly body posture and slow locomotion (*Figure 3D*). Expression of LGC-46(M314I) under its own promoter in *lgc-46(0)* background mimicked the locomotion phenotypes, indicating that LGC-46(M314I) is responsible for the phenotype. We systematically examined null mutations in seven other ACC genes (*acc-1* to *acc-4* and *lgc-47* to *lgc-49*) and observed that all single mutants showed grossly normal movement. We then made double mutants with *lgc-46(gf),* and found that *acc-4(0)* showed a specific suppression of the *lgc-46(gf)* phenotype such that double mutant animals regained normal locomotion (*Figure 5—figure supplement 1A*). *acc-4* was also required for the suppression of *acr-2 (gf)* convulsion by *lgc-46(gf)* (*Figure 5A*). ACC-4 is expressed in all cholinergic motor neurons (*Pereira et al., 2015*). Transgenic expression of *acc-4,* driven by its own promoter, or of an *acc-4* cDNA in cholinergic motor neurons rescued the effects of *acc-4(0)* in *lgc-46(gf) acc-4(0); acr-2(gf)* background (*Figure 5A*), indicating that both ACC-4 and LGC-46 function in the cholinergic motor neurons.

To examine the localization of ACC-4, we generated a functional GFP-tagged ACC-4 using a chimeric approach, as direct tagging to the intracellular loop of ACC-4 impaired its function. We replaced the ACC-4 intracellular loop 3 with GFP-inserted intracellular loop 3 of LGC-46 (*Figure 5— figure supplement 1B*). The resulting ACC-4::GFP protein produced a functional subunit, as its expression in cholinergic motor neurons resulted in suppression on the convulsion frequency of *lgc-*

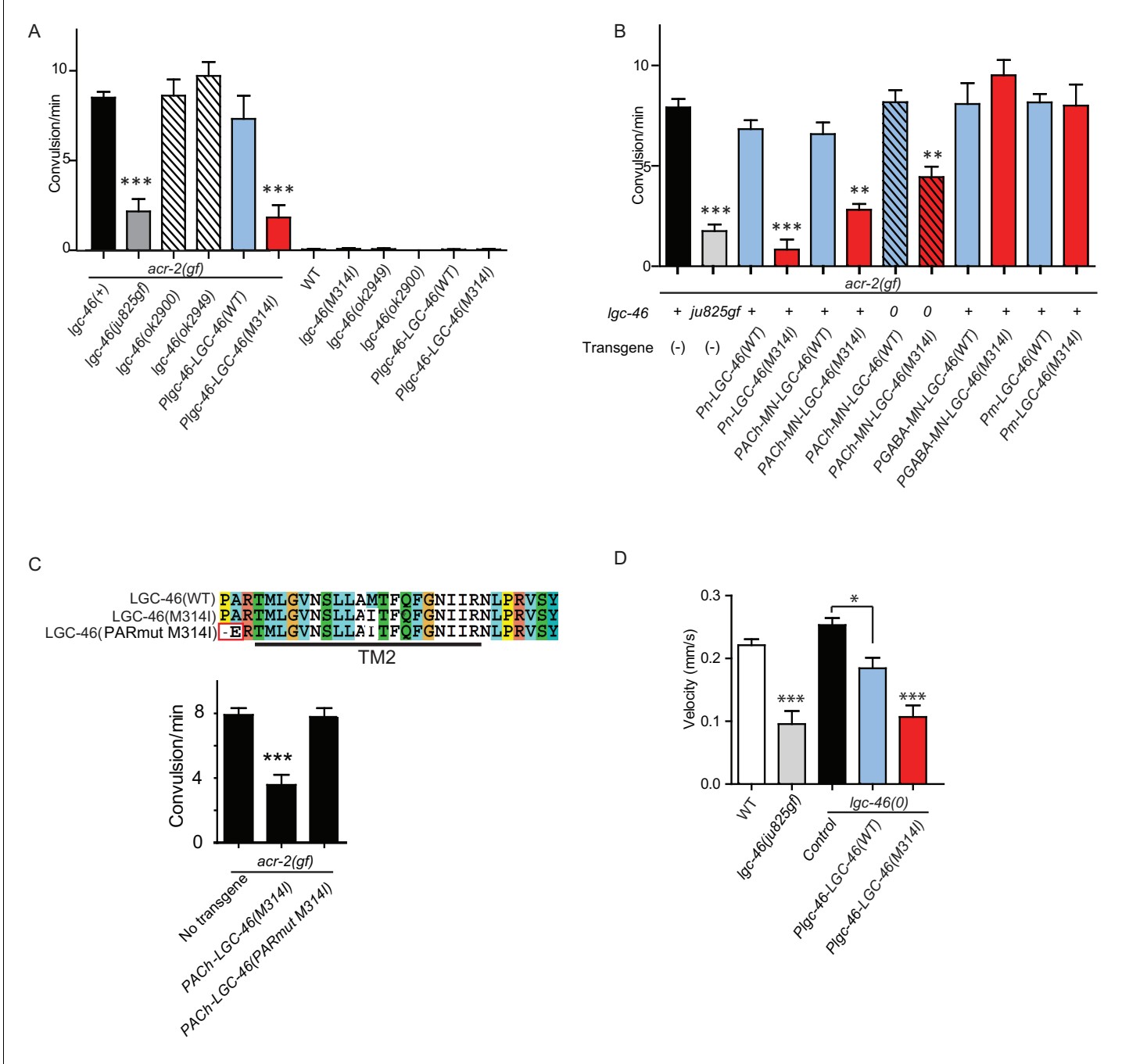

**Figure 3.** A. gain-of-function mutation LGC-46(M314I) affects excitation and inhibition imbalance in locomotor circuit. (**A**) Quantification of convulsion frequencies. *lgc-46(ju825)*, but not *lgc-46(0)*, suppresses *acr-2(gf)* convulsions. Overexpression of LGC-46(M314I) under its own promoter also strongly suppresses convulsion. Data shown as mean ± SEM; n ≥ 16. (**B**) Overexpression of LGC-46(M314I) in cholinergic motor neurons can suppress *acr-2(gf)* convulsions. Data shown as mean ± SEM; n ≥ 16. Promoters used to drive expression are, *Prgef-1* for neurons, *Punc-17β* for cholinergic motor neurons, *Punc-25* for GABAergic motor neurons, *Pmyo-3* for muscles (See **Supplementary file 1,2**). (**A** and **B**) Statistics: one-way ANOVA followed by Dunnett's post-hoc test. **p<0.01, ***p<0.001. (**C**) (Top) Amino acid sequences of LGC-46 wild type, M314I, and PAR motif mutant M314I. (Bottom) Convulsion frequencies of each genotype are shown. Cholinergic motor neuron-specific expression of LGC-46(M314I) suppresses *acr-2(gf)* convulsions, whereas the PAR motif mutant LGC-46(P301Δ A302E M314I) does not. Data shown as mean ± SEM; n = 24, 16, 16, respectively. Statistics: one way ANOVA followed by Dunnet's post-hoc test. *p<0.05, ***p<0.001. (**D**) Off-food velocities for each genotype. Statistics: one-way ANOVA followed by Bonferroni's post-hoc test. Data shown as mean ± SEM. **p<0.01, ***p<0.001.

The following figure supplement is available for figure 3:

*Figure 3 continued on next page*

*Figure 3 continued*

**Figure supplement 1.** Pharmacological assays of aldicarb and levamisole.

46(gf); acc-4(0); acr-2(gf) (**Figure 5A**). ACC-4::GFP showed axonal punctate localization at presynaptic terminals (**Figure 5B–D**), similar to LGC-46::GFP. acc-4(0) did not affect the punctate localization of LGC-46 (**Figure 5E**). These results suggest that ACC-4 and LGC-46 are independently trafficked to the presynaptic site.

We further examined how ACC-4 regulates the SV release by electrophysiological recordings. acc-4(0) single mutants showed normal amplitudes of eEPSCs, but significantly increased eEPSC amplitudes in lgc-46(gf) background (**Figure 5F–H**), indicating that loss of ACC-4 abolished the gain-of-function effects of LGC-46(M314I). The time constants of the rise phase were not affected in acc-4(0) (**Figure 5I**). However, acc-4(0) prolonged the decay phase in wild type or in lgc-46(gf) background, similar to lgc-46(0) (**Figure 5I**). Together, these data support a conclusion that LGC-46 and ACC-4 function together in cholinergic motor neurons to regulate the SV release kinetics following neuronal activation. As LGC-46 and ACC-4 both belong to the ACC protein family, it is possible that the proteins may form a heteromeric channel. Alternatively, they may interact indirectly and co-regulate cholinergic release. Future experiments will determine their mode of interaction, as well as other components and channel property of this LGIC.

## Discussion

The notion of presynaptically acting LGICs in synaptic transmission is long known, with extensive evidence primarily from classical electrophysiological recordings and pharmacology. However, few previous studies have addressed subcellular localization of anionic LGICs in presynaptic release sites (*Belenky et al., 2003*; *Hruskova et al., 2012*; *Sur et al., 1995*), partly because of limitations in available reagents and in detection methodology to directly examine the function and localization of presynaptic LGICs. Here, using genetic, molecular, cell biological and physiological tools in *C. elegans*, we have provided multiple lines of evidence to demonstrate that a presynaptic anion-selective ion channel acts as an auto-receptor to ensure tight control of neurotransmitter release in the motor circuit. LGC-46 localizes to the presynaptic release sites in cholinergic motor neurons. Loss of lgc-46 specifically affects evoked SV release in the late phase, without affecting endogenous SV release. A gain-of-function mutation LGC-46(M314I) affects both endogenous and evoked SV release. As both loss- and gain- of function lgc-46 mutants specifically affect the decay phase, but not the rise phase, of eEPSC, our data support a model where upon a burst of ACh release, opening of LGC-46 channels limits further SV release in the sustained late phase. LGC-46(gf) also exhibits a strong inhibitory effect on synaptic release under elevated neuronal activity, suggesting that the mutant channel could have increased sensitivity to ligands, higher channel conductance or prolonged open state, thereby affecting the basal activity of the cholinergic motor neurons.

Rapid feedback inhibition of SV release upon stimulation can contribute to temporal resolution of neural activity within the circuit. A tightly regulated sharp decay of evoked EPSCs can ensure transient and adequate release of neurotransmitter to match high-frequency stimuli, possibly without triggering desensitization of postsynaptic receptors. Sharpening of synaptic activities by presynaptic LGICs can minimize unwanted cross-talks between synapses and neurons, allowing accurate information encoding in high frequency. Such mode of neural control can contribute to the overall circuit performance and may be particularly suited for synapses with graded potential, such as the case in *C. elegans* and those in the retina of vertebrates.

Acetylcholine is well known for its action to trigger fast excitatory response through cation channels and to modulate neuronal activity via metabotropic receptors. However, classical electrophysiological studies from *Aplysia* have also shown that acetylcholine can induce inhibitory responses. Injection of acetylcholine to pleural ganglia caused rapid inhibitory postsynaptic potential in a chloride concentration-dependent manner, suggesting the presence of ACh-gated anion channels (*Kehoe, 1972a*, *1972b*; *Kehoe and McIntosh, 1998*). The identity of these channels in *Aplysia* remains unknown. The discovery of *C. elegans* ACC proteins has provided molecular evidence for ACh-gated chloride channels (*Putrenko et al., 2005*). Our understanding of ACC channel properties

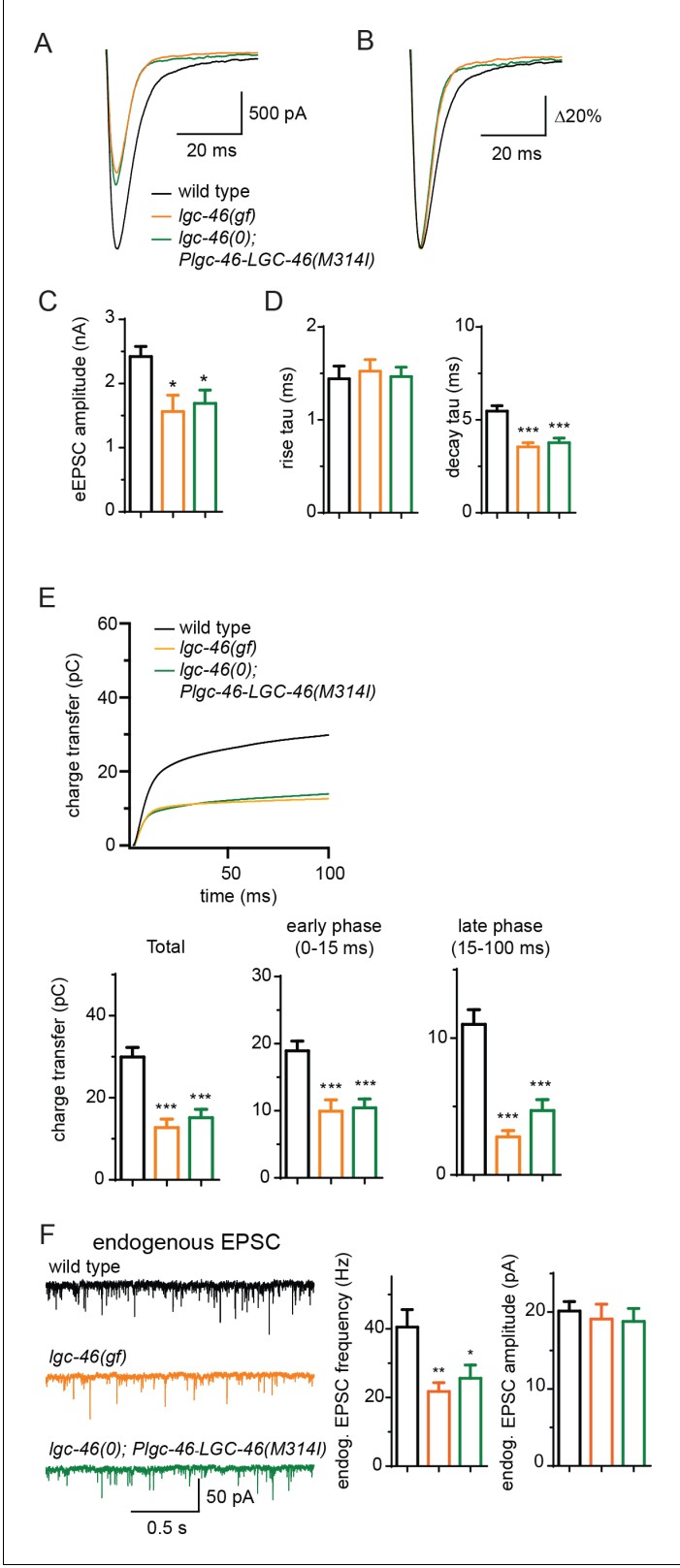

**Figure 4.** LGC-46(M314I) limits synaptic transmission by shortening the decay phase of evoked release. (*A* and *B*) Average traces (*A*) and normalized average traces (*B*) of eEPSCs. (*C*) Mean amplitude of eEPSCs. (*D*) Rise tau and decay tau of eEPSCs. (*E*) Average traces of cumulative charges of eEPSCs, and charge transfers after stimulation. (*F*) Representative traces, mean frequency and mean amplitude of endogenous EPSCs. wild type (n = 12), *lgc-46*

*Figure 4 continued on next page*

*Figure 4 continued*

*(ju825gf)* (n = 10), and *lgc-46(0);Plgc-46-LGC-46(M314I)* (n = 12). Animals were recorded at 20 degrees in 1.2 mM Ca$^{2+}$ bath solutions. Statistics, one-way ANOVA, Bonferroni's post hoc test. ***p<0.001, **p<0.01, *p<0.05. Error bars indicate SEM.

The following figure supplement is available for figure 4:

**Figure supplement 1.** LGC-46(M314I) shows a presynaptic punctate localization pattern similar to LGC-46(WT), and amplitude and kinetics of endogenous EPSCs are normal in *lgc-46(gf)*.

and their in vivo functions is only beginning. Several ACC channels are recently shown to display the differential expression in neurons (*Pereira et al., 2015*). ACC-2 has been implicated in regulation of reversal behavior, presumably as a postsynaptic receptor (*Li et al., 2014*). Our data support a conclusion that ACC-4 acts as a functional partner of LGC-46 at the presynaptic release site. Increasing presynaptic inhibition using *lgc-46(gf)* dampens overexcitation in the *acr-2(gf)* seizure model. The molecular nature and functional properties of *acr-2(gf)* show similarities to gain of function mutations in human CHRNB2, the β2 subunit of neuronal nicotinic receptors, which have been identified as causal in autosomal dominant nocturnal frontal lobe epilepsy (*Phillips et al., 2001*). Studies using cell lines and transgenic mouse models show that these mutant CHRNB2 proteins result in hyperactive channels, and can alter the local circuit activity (*Manfredi et al., 2009*). Our observations that the hyper-excitation in *acr-2(gf)* animals can be suppressed by increasing presynaptic inhibition suggest alternative ideas to manage circuit dysfunction. Growing evidence highlights the importance of presynaptic anion LGICs. For example, malfunction of presynaptic glycine channel can partially contribute to startle disease (hyperekplexia) in a mouse model (*Xiong et al., 2014*). These findings suggest ionotropic anion channels should be considered potential targets for modulating neuronal circuit function and in treatments for neurological disorders.

# Materials and methods

## *C. elegans* genetics, transgenes and strains

*C. elegans* strains were grown at 22.5°C following standard procedures. *Supplementary file 1* lists all the strains and transgene information. Whole-genome sequence analysis of CZ21292 *lgc-46 (ju825); acr-2(gf)* was performed using Galaxy platform (*Giardine et al., 2005*). Molecular biology was performed following standard methods. Gateway recombination technology (Invitrogen, CA) was used for generating expression constructs. *Supplementary file 2* describes the details of DNA constructs generated in this study. The promoters used are: Prgef-1 for pan-neuronal (*Altun-Gultekin et al., 2001*), Pmyo-3 for body muscles (*Okkema et al., 1993*). Pitr-1 for DA9 neuron (*Klassen and Shen, 2007*).

## Molecular biology, RNA analyses and transgenes

High-copy transgenic arrays were generated following standard protocols. Single-copy insertion lines of *Punc-17β-LGC-46(WT)::GFP* and *Punc-17β-LGC-46(M314I)::GFP* were made at cxTi10882 site on Chromosome IV (*Frøkjær-Jensen et al., 2012*) using modified vectors. Briefly, N2 animals were injected with following three constructs: a construct carrying the *Punc-17β-LGC-46(WT)::GFP* sequence with cxTi10882 homology arms and a copy of hygromycin resistance gene; a construct which drives expression in the germline of Cas9 and sgRNA targeted for the cxTi10882 region, and a fluorescent coinjection marker. F2 animals resistant to hygromycin were selected. Loss of the fluorescent coinjection marker indicated the loss of extrachromosomal array. Insertion of single copy in the genomic locus was confirmed by PCR using primers outside of the homology arms. The single copy insertion strains were outcrossed twice before use in further experiments. Generally, we found that expression levels of transgenes using the *Punc-17β* promoter to be comparable to those by *Plgc-46.*

To verify *lgc-46* transcripts in wild type and mutants, we isolated mRNA from animals of mixed stages using Trizol (ThermoFisher Scientific), and generated cDNAs using SuperScript III

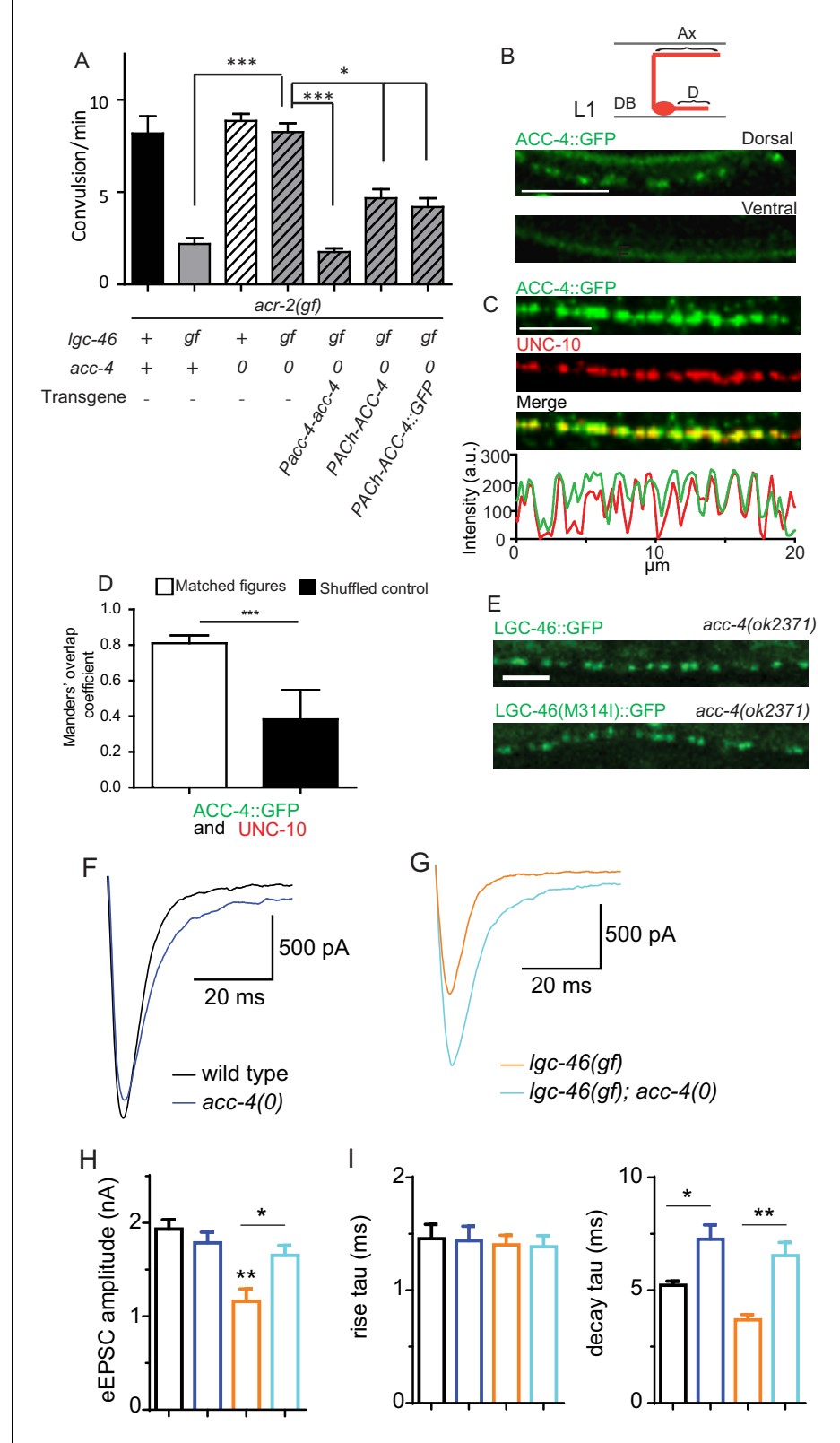

**Figure 5.** ACC-4 is required for the function of LGC-46(M314I), and localizes presynaptically. (**A**) Frequency of convulsion in animals of indicated genotypes. Loss of function in *acc-4* reverses the suppression effect by *lgc-46 (ju825)* on *acr-2(gf)*. The expression of *acc-4(+)* under the endogenous promoter or cholinergic motor neuron-specific promoter can rescue the effect. Data shown as mean ± SEM; n ≥ 16. Statistics: one way ANOVA followed

*Figure 5 continued on next page*

*Figure 5 continued*

by Dunnet's post-hoc test. **p<0.01, ***p<0.001. (BD) ACC-4::GFP localizes to presynaptic terminals of cholinergic motor neurons, similar to LGC-46::GFP. Images are from *lgc-46(gf) acc-4(0); acr-2(gf); Punc-17b-acc-4::GFP (juEx7438)*. (B) Upper panel shows a schematic of a cholinergic motor neuron in L1 animals. The lower panel shows a confocal image of ACC-4::GFP showing punctate localization in the dorsal nerve cord. (C) ACC-4::GFP colocalizes with active zone protein UNC-10/RIM. (D) Mander's overlap coefficient showing the extent of pixel colocalization of ACC-4 and UNC-10 (open bar) from immunostaining. As negative controls, Mander's overlap coefficient from shuffled images are shown (Filled bar). For each genotype, 10 images were analyzed. Data shown as mean ± SD. Statistics: Student's t-test. ***p<0.001. (E) The punctate localization of LGC-46 is maintained in the *acc-4(ok2371)* null mutants. (*F* and *G*) Average traces of evoked release (eEPSC) from wild type (n = 10), *acc-4(0)* (n = 10), *lgc-46(gf)* (n = 10), and *lgc-46(gf);acc-4(0)* (n = 10). (H) Mean amplitude of eEPSC. (I) Rise tau and decay tau of eEPSC. Animals were recorded at 17°C in 2 mM Ca$^+$ bath solutions. Statistics, one-way ANOVA. Bonferroni's post hoc test for eEPSC comparisons. **p<0.01; *p<0.05. Error bars indicate SEM.

The following figure supplement is available for figure 5:

**Figure supplement 1.** ACC-4 is required for the function of LGC-46(M314I).

---

(ThermoFisher Scientific), following the manufacturer's protocol. *lgc-46* cDNA was amplified using following primers:

```
YJ11741: 5'ATGCAATATCTGCAATTCCT3'
```

```
YJ11742: 5'TTATATTTATTATCATTCGTTGACTAG3'.
```

Sequence analyses of cDNAs from *ok2949* and *ok2900* showed neither allele would produce any functional protein. *lgc-46* cDNAs from N2 and *lgc-46(ju825); acr-2(n2420)* were cloned into Topo PCR8 vector (Invitrogen). *acc-4* cDNA from N2 was amplified using following primers:

```
YJ11811: 5'ATGCGACTAATCATATTAGTAATCTCCATTC3'
```

```
YJ11812: 5'CTTAGATAGTTCTAACCAATAGTTTTCCGAG3'
```

The amplified fragment was cloned into Topo PCR8 vector.

## Locomotion, convulsion behavior, and pharmacology analyses

For video recording of locomotion, L4 larvae were transferred to NGM plates seeded with OP50. On the next day, young adults were transferred to fresh plates without OP50, allowed to crawl away from bacteria, and were then transferred to assay plates without OP50. Videos of the animals were captured for 5 min at 3 fps using Pixelink camera and analyzed using a multi-worm tracking custom software. Average velocities of 10 animals per strain were obtained; and the experiment was repeated three times per strain. Quantification of convulsion and pharmacological analysis were performed as previously described (*Jospin et al., 2009*; *Stawicki et al., 2013*).

## Immunocytochemistry and imaging of GFP reporters

Whole-mount immunostaining was performed essentially as previously described (*Van Epps et al., 2010*). Confocal images of fluorescent markers were collected under identical settings using Zeiss LSM 710 confocal microscope (63x objective). The adult animals were immobilized using 0.05 µm polystyrene beads (Polysciences) and placed on 10% agarose pads with the dorsal side facing the coverslip. Larvae at the designated stage were anesthetized using 1 mM levamisole and placed on 4% agarose pads. Maximum-intensity Z stack images were obtained from 3 sections at 0.5 µm intervals. Images were processed using ImageJ software. Linescan analyses were performed using MetaMorph (Molecular Devices Corp.). For fluorescent puncta analysis in *Figure 2—figure supplement 1B*, signal intensities obtained from linescan analyses were imported to IGOR Pro (WaveMetrics, Lake Oswego, OR) and processed as previously described (*Zhou et al., 2013*). Linescans from dorsal

nerve cords (20 µm per animal) from six animals per genotype were analyzed. For quantification of colocalization of LGC-46 and presynaptic proteins, 10 images per genotype were processed by ImageJ plugin Intensity Correlation Analysis and Mander's overlap coefficients were obtained (*Manders et al., 1993*). In order to examine whether the correlation occurred by chance, images from red and green channels were shuffled and correlation of pixels between animals were obtained as controls (shuffled control).

## Electrophysiology

Neuromuscular dissection and recording methods were adapted from previous studies (*Richmond et al., 1999*; *Zhou et al., 2013*). The recipes of bath solution and pipette solution were adapted from (*Madison et al., 2005*). Adult worms were immobilized on Sylgard-coated cover slips with cyanoacrylate glue. After dissection, conventional whole-cell recordings from muscle cells were performed.

The bath solution contains (in mM): 127 NaCl, 5 KCl, 26 NaHCO$_3$, 1.25 NaH$_2$PO$_4$, 4 MgCl$_2$, 10 glucose and sucrose to 340 mOsm with indicated CaCl$_2$ concentrations, bubbled with 5% CO$_2$, 95% O$_2$ at 20°C. The pipette solution contains (in mM): 120 CH$_3$O$_3$SCs, 4 CsCl, 15 CsF, 4 MgCl$_2$, 5 EGTA, 0.25 CaCl$_2$, 10 HEPES and 4 Na$_2$ATP, adjusted to pH 7.2 with CsOH. Conventional whole-cell recordings from muscle cells were performed with 2.5–3.5 MΩ pipettes. An EPC-10 patch-clamp amplifier was used together with the Patchmaster software package (HEKA Electronics, Lambrecht, Germany). Endogenous EPSCs were recorded at −60 mV. For recording evoked EPSCs, a second glass pipet filled with bath solutions was put on the ventral nerve cord as stimulating electrode. The stimulating electrode gently touched the anterior region of ventral nerve cord to form loose patch configuration, which is around 1 muscle distance from recording pipets. A 0.5 ms, 85 µA square current pulse was generated by the isolated stimulator (WPI A320RC) as stimulus. Series resistance was compensated to 70% for the evoked EPSC recording. All current traces were imported to IGOR Pro (WaveMetrics, Lake Oswego, OR) for further analysis. A single exponential equation was used to fit the rise phase or decay phase of eEPSCs. The traces for the cumulative transferred charge were obtained by integration of eEPSCs.

## Data analyses and statistics

Statistical analyses were performed using Graphpad Prism v5. One-way ANOVA, Two-way ANOVA, Dunnet's tests, Bonferroni tests were used.

## Acknowledgements

We thank N Pokala, C Bargmann, A Calhoun, S Chalasani for sharing the multi-worm tracking analyses software prior to publication, J Dent for sharing unpublished data on ACC channels, YB Qi for isolating *ju825*, Z Wang for CasCi protocol, and AD Chisholm, J Wang, D Berg, and our lab members for comments. Some strains were obtained from the Japan National BioResource Project (NBRP) and the *Caenorhabditis* Genetics Center (CGC). S T-K was a recipient of the Nakajima Foundation Predoctoral Fellowship. Y J is an Investigator, and K Z, a research associate, of the Howard Hughes Medical Institute.

## Additional information

### Funding

| Funder | Grant reference number | Author |
| --- | --- | --- |
| The Nakajima Foundation | Predoctoral Fellowship | Seika Takayanagi-Kiya |
| Howard Hughes Medical Institute | | Yishi Jin |
| National Institutes of Health | R01 NS035546 | Yishi Jin |

The funders had no role in study design, data collection and interpretation, or the decision to submit the work for publication.

## Author contributions

ST-K, KZ, Conception and design, Acquisition of data, Analysis and interpretation of data, Drafting or revising the article; YJ, Conception and design, Analysis and interpretation of data, Drafting or revising the article

## Author ORCIDs

Yishi Jin, http://orcid.org/0000-0002-9371-9860

## Additional files

### Supplementary files

- Supplementary file 1. List of strains used in the study.
- Supplementary file 2. List of constructs used in the study.

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
