## [Decision Letter]

Thank you for submitting your article "Release-dependent feedback inhibition by a presynaptically localized ligand-gated anion channel" for consideration by *eLife*. Your article has been reviewed by two peer reviewers, and the evaluation has been overseen by Oliver Hobert as Reviewing Editor and Eve Marder as the Senior Editor. The following individuals involved in review of your submission have agreed to reveal their identity: Jeremy S Dittman (Reviewer #1); Shawn Xu (Reviewer #2).

You will see the complete reviews at the end of this message. After discussion among the reviewers and reviewing editor, we decided that only minimal revisions are required for this manuscript:

1) While the ACC-4 data should be kept in the manuscript as is, we would like to see the caveats of the present analysis pointed out at the very end of the last paragraph of the Result section. Those caveats are nicely summarized by reviewer #2 and should be mentioned.

2) Figure 2 and Figure 4 should include summary graphs of the endogenous EPSC amplitude in addition to the frequency data shown.

Reviewer #1:

This is a nice study using a combination of genetics and electrophysiology to explore the physiological role of a presynaptic ligand-gated ion channel using *C. elegans*. The authors investigate LGC-46, which is likely to be a chloride channel gated by ACh (the major neuromuscular junction transmitter) based on sequence homology. Based mainly on a gain of function mutation in the lgc-46 gene, the authors conclude that synapses are normally inhibited when high levels of activation elevate ACh and open a presynaptic chloride conductance (ie a negative feedback loop). Strengths of this work include nice imaging to demonstrate presynaptic localization, beautiful electrophysiology of the NMJ, and solid genetics. The weaknesses are mainly that there does not seem to be much impact on the animal when LGC-46 is removed (perhaps redundancy issues) and a basic characterization of the LGC-46 channel properties is lacking. Although the lack of a pronounced behavioral phenotype in the null mutant (and subtle electrophysiological phenotype) may be considered a serious drawback to this study, the gain-of-function experiments still provide compelling and interesting insights into the biology of *lgc-46*. In fact, if this is a redundant regulatory system at the synapse, the gain-of-function approach is a more informative one until all of the players are identified. Of perhaps more concern, there is no functional characterization of the *lgc-46* gene product. Much of the interpretation for the gain-of-function M314I mutation and the PAR motif mutations in LGC-46 would be stronger if we knew that LGC-46 is truly an ACh-gated chloride channel and that the point mutations do what we think they do. For example, this has been done for the worm ACC subunits (Dent lab) and other LGC subunits (Horwitz lab). Is it okay to rely on studies of homologous ionotropic receptors? It's probably not fair or reasonable to ask for everything in one study. That concern aside, I do not have any major concerns about the technical quality of the data and I think that the development of a genetic approach to understanding the role of inhibitory ionotropic receptors in the presynaptic terminal is a significant advance in the field. I support going forward with this manuscript.

Reviewer #2:

The role of LGICs in SV release has most been characterized postsynaptically. Here, the authors demonstrate a novel role for ACh-gated chloride channels which act at the presynaptic site to negatively inhibit synaptic vesicle release. They record evoked and spontaneous EPSCs from in vivo muscle cells to show that LGC-46, an ACh-gated anion-selective channel, accelerates the decay phase of eEPSCs. LGC-46 colocalizes to presynaptic markers and loss-of-function mutations are hypersensitive to aldicarb but not levamisole, indicating presynaptic function. Gain-of-function mutations accelerate the decay phase of the EPSC and reduces severity of an ACh-related over-excitation behavioral phenotype. The authors also show that ACC-4 localizes to presynaptic terminals and fund that ACC-4 may form a heteromeric channel with LGC-46. ACC-4 is also shown to regulate the evoked release kinetics of synaptic vesicles at cholinergic motor neuron synapses. Overall, this is an interesting study and will significantly advance our understanding of how LGICs presynaptically modulate SV release at the molecular and cellular levels.

The authors present strong evidence supporting that ACC-4 and LGC-46 function together. The simplest explanation is that they function as a heteromeric channel because they both are pore-forming subunits (not auxiliary subunit-like), but the authors did not provide any direct evidence supporting one way or the other. Instead, they simply avoid touching on this notion, but the readers will not. This leaves behind a series of unanswered questions. For example, do they physically interact? Do they form a functional channel when expressed in a heterologous system? I understand that the authors may not have the expertise to perform such experiments. If so, my suggestion is to remove ACC-4 data. The data on LGC-46 may stand on its own. This would not affect the integrity of the paper.

Figure 2 and Figure 4 should include summary graphs of the endogenous EPSC amplitude in addition to the frequency data shown.

---

## [Author Response]

You will see the complete reviews at the end of this message. After discussion among the reviewers and reviewing editor, we decided that only minimal revisions are required for this manuscript:

1) While the ACC-4 data should be kept in the manuscript as is, we would like to see the caveats of the present analysis pointed out at the very end of the last paragraph of the Result section. Those caveats are nicely summarized by reviewer #2 and should be mentioned.

2) Figure 2 and Figure 4 should include summary graphs of the endogenous EPSC amplitude in addition to the frequency data shown.

First, we greatly appreciate the reviewers’ comments and will take their suggestions in our future experiments.

We have revised the manuscript to incorporate reviewer 2’s caution on roles of ACC-4. “As LGC-46 and ACC-4 both belong to the ACC protein family, it is possible that the proteins may form a heteromeric channel. […] Alternatively, they may interact indirectly and co-regulate cholinergic release. Future experiments will determine their mode of interaction, as well as additional components and the channel property of this LGIC”

Second, the summary graphs of the endogenous EPSC amplitude were in the supplementary figure for Figure 2 and Figure 4. We now moved these graphs to the main Figure 2 and Figure 4, and revised the figure legends.